# Small molecule inhibitors and a kinase-dead expressing mouse model demonstrate that the kinase activity of Chk1 is essential for mouse embryos and cancer cells

Somsundar V Muralidharan* , Lisa M Nilsson* , Mattias F Lindberg , Jonas A Nilsson

Chk1 kinase is downstream of the ATR kinase in the sensing of improper replication. Previous cell culture studies have demonstrated that Chk1 is essential for replication. Indeed, Chk1 inhibitors are efficacious against tumors with high-level replication stress such as Myc-induced lymphoma cells. Treatment with Chk1 inhibitors also combines well with certain chemotherapeutic drugs, and effects associate with the induction of DNA damage and reduction of Chk1 protein levels. Most studies of Chk1 function have relied on the use of inhibitors. Whether or not a mouse or cancer cells could survive if a kinase-dead form of Chk1 is expressed has not been investigated before. Here, we generate a mouse model that expresses a kinase-dead (D130A) allele in the mouse germ line. We find that this mouse is overtly normal and does not have problems with erythropoiesis with aging as previously been shown for a mouse expressing one null allele. However, similar to a null allele, homozygous kinase-dead mice cannot be generated, and timed pregnancies of heterozygous mice suggest lethality of homozygous blastocysts at around the time of implantation. By breeding the kinase-dead Chk1 mouse with a conditional allele, we are able to demonstrate that expression of only one kinase-dead allele, but no wild-type allele, of *Chek1* is lethal for Myc-induced cancer cells. Finally, treatment of melanoma cells with tumor-infiltrating T cells or CAR-T cells is effective even if Chk1 is inhibited, suggesting that Chk1 inhibitors can be safely administered in patients where immunotherapy is an essential component of the arsenal against cancer.

## Introduction

Replication is tightly controlled to ensure accurate copying of the large DNA molecule [1]. Not only the initiation of replication but also the progression of the replication fork is regulated by kinases [2]. If the cell experiences replication stress, UV-induced DNA damage, and enzymatic and helicase remodeling after DNA inter-strand cross-linking chemotherapy or environmental agents, the replication fork is stalled [3]. This causes single-strand DNA that induces accumulation of replication protein A which recruits the ATR kinase [4, 5, 6]. ATR phosphorylates Chk1 kinase and other proteins to ensure replication is being blocked as to avoid replication fork collapse and DNA damage [7, 8, 9]. Both ATR and Chk1 are regarded essential kinases [10, 11, 12], albeit Chk1-deficient fission yeast and chicken lymphoma cells can be generated [13, 14]. These display defects in DNA damage response and mitosis because Chk1 also have a role there [15].

Oncogenes such as Myc and Ras can cause replication stress [16], which makes cancer cells sensitive to ATR or Chk1 inhibition [17, 18, 19, 20, 21, 22]. Insertion of an extra copy of the *Chek1* gene in mice protects cells from replication stress, prolongs survival in *Atr* hypomorphic (Seckel) mice, and facilitates transformation [23]. Furthermore, we previously showed that Chk1 levels are elevated in Myc-induced cancer cells and both genetic and pharmacological inhibition of Chk1 resulted in the cell death of cancer cells in vivo with no evident toxicity in mice [18]. These data show that there may be a therapeutic window targeting ATR and Chk1, despite they are essential.

A surprising effect of Chk1 inhibition in Myc-induced lymphoma cells was also the fact that the protein levels of Chk1 were reduced in Chk1 inhibitor–treated cancer cells [18]. Because of this, an outstanding question remained whether or not the effect of Chk1 inhibitors was only due to inhibition of Chk1 kinase activity or a lack of the Chk1 protein, or both. To model this, and the importance of Chk1 activity in embryogenesis and tumorigenesis, we here describe the phenotypes of Chk1 kinase-dead mice. We also investigate the role of Chek1 in lymphoma, in erythropoiesis and in T-cell–mediated tumor cell killing, using inhibitors and/or mice expressing a kinase-dead *Chek1* allele.

## Results

### Chk1 kinase activity is essential for embryogenesis

We have previously shown that the highly selective Chk1 inhibitor Chekin can kill Myc-induced lymphoma cells [18]. This coincides

Department of Surgery, Sahlgrenska Cancer Center, Institute of Clinical Sciences at University of Gothenburg, Gothenburg, Sweden

Correspondence: jonas.a.nilsson@surgery.gu.se
*Somsundar V Muralidharan and Lisa M Nilsson contributed equally to this work

with an induction of DNA damage and a reduction in Chk1 levels as assessed by immunoblotting. To investigate if these effects can be reproduced using Chk1 inhibitors under clinical development, we repeated these experiments using LY2603618 and CCT245737. Both compounds induced DNA damage and cell death, which coincided with a reduction in Chk1 levels (Fig 1A and B). We also observed that levels of phosphorylated Chk1 increased at serine 345, and this phosphorylation appears to be mediated by ATR because the ATR inhibitor VE821 ameliorated the S345 phosphorylation induced by Chk1 inhibitor (Fig 1C). Small molecule inhibitors can potentially have off-target effects because their small size results in that they fit many pockets of proteins (24). They can also be pumped out of cells or remain inaccessible for tissues and developing embryos (25). To model Chk1 inhibition in vivo, we generated a mouse carrying a point mutation in the *Chek1* gene which is calculated to result in a substitution of aspartate 130 into an alanine in the mouse Chk1. This aspartate is essential for the kinase activity of Chk1, resides in a highly homologous region of the protein (Fig 2A), and a substitution to an alanine creates a kinase-dead form of Chk1 (26). The targeting vector (Fig 2B) used was transfected into mouse C57BL/6-ES, cells and correct targeting was confirmed by PCR and Southern blot. Two correctly targeted ES cell clones were injected into blastocysts, and several high chimera founders were born. These appeared normal and breeding to C57BL/6 mice generated offspring that were heterozygous or wild-type for the *Chek1*[D130A] allele (Fig 2C) and that were also normal. Generation of MEFs from wild-type and heterozygous *Chek1*[D130A/wt] embryos resulted in growth patterns that were similar irrespective of the genotype (Fig S1A and B). Interbreeding between heterozygous *Chek1*[D130A] mice generated a non-Mendelian ratio of offspring. No mice were born that were homozygous for the *Chek1*[D130A] allele (Fig 2D), suggesting that the kinase activity of Chk1 is essential for embryogenesis. To verify that the PCR strategy was correct, we performed timed pregnancies between heterozygous *Chek1*[D130A] mice and genotyped embryos at different times post-coitus. The only stage where we were able to detect homozygous *Chek1*[D130A] embryos were at embryonic day 3.5 (E3.5; Fig 2D). The amount of heterozygous *Chek1*[D130A] embryos at E10.5 and at birth was not at a Mendelian ratio (Fig 2D, Fisher's test *P* = 0.0162), which was surprising because

breeding of heterozygous mice to wild-type C57BL/6 mice generated heterozygous offspring in a Mendelian frequency (Fig 2E).

## Half the level of Chk1 kinase activity does not induce anemia in aged mice

A previous study using heterozygous *Chek1* knockout mice demonstrated that *Chek1* appeared to be "haploinsufficient" for red blood cell production (27). To investigate if it was the kinase activity or the Chk1 protein that constituted this phenotype, we aged *Chek1*[D130A/wt] mice for 1 yr and monitored their blood status. Surprisingly, aged mice from both lines of mice carrying one kinase-dead copy of Chk1 displayed normal values on all parameters analyzed. We assessed hemoglobin (Hb, Fig 3A), RBCs (Fig 3B), and platelets (PLT, Fig 3C) on mice 45–60 wk of age, both females and males. In addition, we compared spleen weights of the same cohorts by ages 50–65 wk, and also here the groups are indistinguishable from each other (Fig 3D). Boles et al describe specific defects during erythropoiesis in *Chek1* heterozygotous mice. To gain more insight into erythropoiesis in our mice carrying one copy of kinase-dead Chk1, we stained cells isolated from the bone marrow of our aged mice and subjected these to staining with Ter119-APC and CD71-PE and subsequent flow cytometry analysis. Also in this assay, we fail to detect any differences between the two groups (Figs 3E and S2).

## Chk1 kinase activity is essential for cell survival

Since kinase-dead Chk1 mice cannot be generated by interbreeding, we obtained *Chek1* conditional knockout mice (*Chek1*[Flox/Flox]) and mice expressing a fusion between the Cre recombinase and the ligand-binding domain of the estrogen receptor (*CreER* mice). By breeding these mice to heterozygous kinase-dead Chk1 mice, we would be able to delete the wildtype allele of *Chek1* by Cre activation with the estrogen analog Tamoxifen after the mice were born, thereby leaving only the kinase-dead allele to be expressed. Administration of tamoxifen to *CreER;Chek1*[Flox/D130A] mice, but not to *CreER;Chek1*[Flox/wt] mice resulted in toxicity (data not shown) and the ethical limitations prohibited us from further experiments using

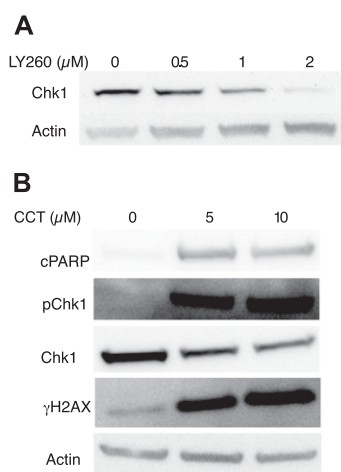

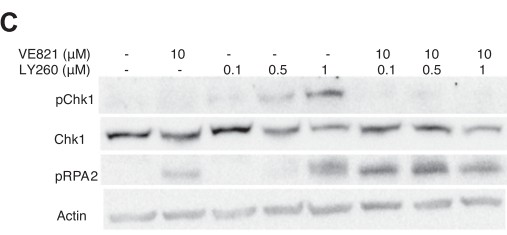

**Figure 1. Chk1 inhibitors reduce expression of Chk1 and induce DNA damage.**
**(A)** λ820 mouse lymphoma cells were treated with LY2603618 for 24 h, and cells were analyzed by immunoblotting for Chk1 and loading control β-actin. **(B)** λ820 mouse lymphoma cells were treated with CCT245737 and cells were analyzed by immunoblotting for apoptosis marker cleaved PARP, total or phosphorylated (p)Chk1 and DNA damage marker phosphorylated histone 2AX (γH2Ax). **(C)** λ820 mouse lymphoma cells were treated with LY2603618 or ATR inhibitor VE821 for 24 h, and cells were analyzed by immunoblotting for total Chk1, pChk1, phosphorylated (p)RPA2, and loading control β-actin.
Source data are available for this figure.

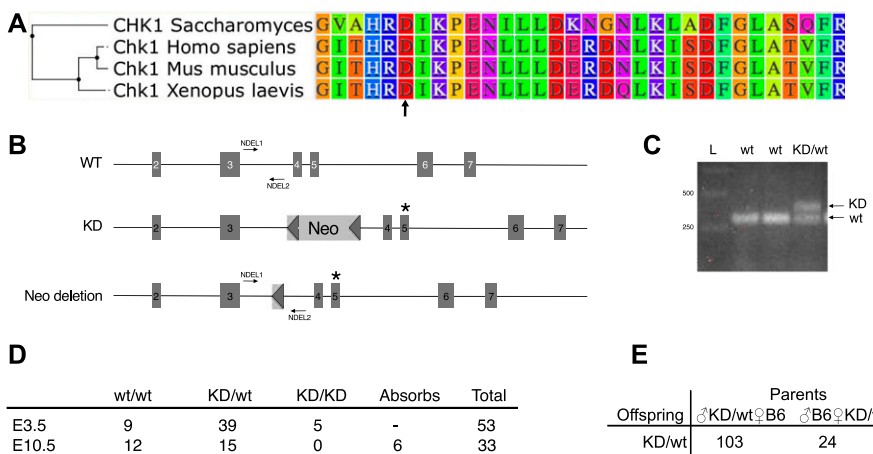

**Figure 2. Generation of a kinase-dead Chk1 mouse model.**
**(A)** Clustering analysis of Chk1 from yeast, human, mouse, and frog. Arrow indicated aspartate 130 (D130). **(B)** Targeting strategy to generate a kinase-dead *Chek1* allele. *Top* part of the mouse *Chek1* gene. *Middle* part of the Chek1 gene showing targeted allele in ES cells. *Bottom* part of the *Chek1* gene showing the *Chek1* gene after removal of neo cassette in an *Flp*-deleted mouse. **(C)** PCR genotyping of three offspring after mating between a heterozygous *Chek1* knockin mouse (*Chek1*^D130A/wt^). **(B)** PCR primers (NDEL1 and 2) are shown in (B). **(D)** Results of genotyping of embryos or offspring from time pregnancies between *Chek1*^D130A/wt^ females and *Chek1*^D130A/wt^ males. **(E)** Results of genotyping of offspring from crosses of heterozygous *Chek1* knockin males or females to C57BL/6 mice.

this system. We therefore interbred mice carrying *CreER* and floxed or kinase-dead *Chek1* alleles with *Cdkn2a*$^{-/-}$ mice and generated mouse fibroblasts from interbreedings. We could then investigate the effect of deletion of the wild-type allele of cultured cells because *Cdkn2a*$^{-/-}$ mouse embryo fibroblasts do not undergo culture-induced senescence. Addition of the estrogen analog 4-hydroxy-tamoxifen (4-HT) to MEFs of different genotypes demonstrated that MEFs relying on just a kinase-dead form of Chk1 triggers DNA damage as revealed by phosphorylation of H2AX (γH2AX, Fig 4A). The induction of γH2AX upon loss of active Chk1 was equal to that of UV irradiation. To investigate if cells expressing oncogenes such as Myc would be sensitive to a kinase-dead Chk1 allele, we transduced *CreER*; *Chek1*^Flox/D130A^;*Cdkn2a*$^{-/-}$ or *CreER*;*Chek1*^Flox/wt^;*Cdkn2a*$^{-/-}$ cells with a retrovirus expressing c-Myc. Cre activation with 4-HT resulted in a loss of viability of Myc-expressing fibroblasts (Fig 4B). This correlated with an induction of replication stress, that is, phosphorylation of

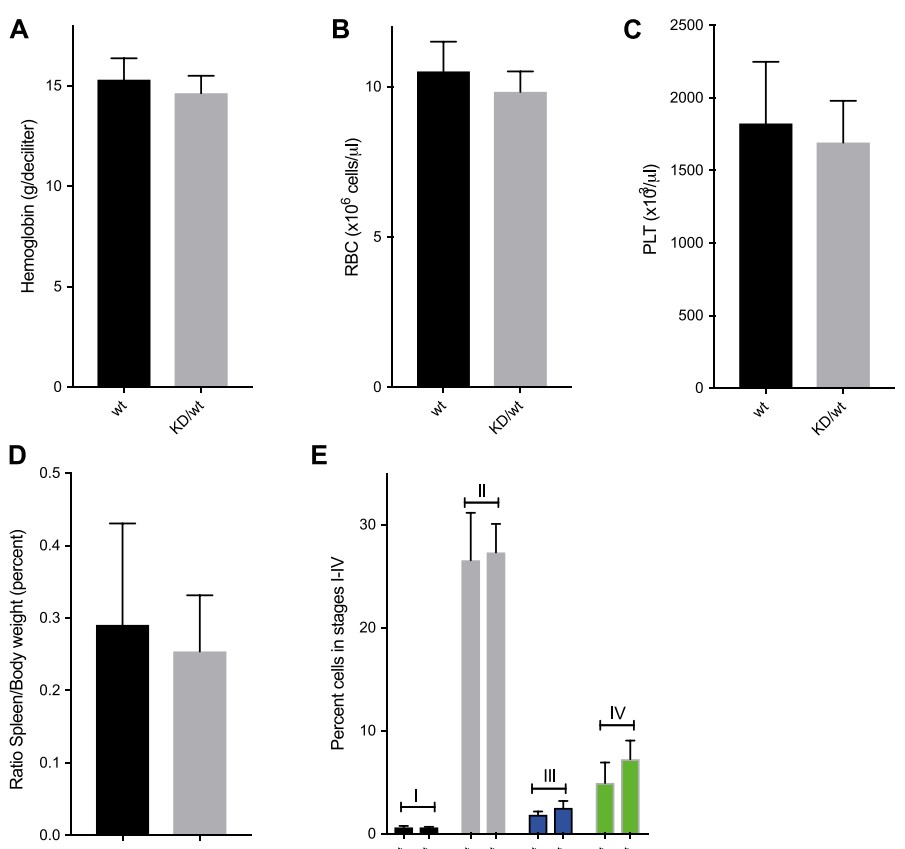

**Figure 3. Heterozygous *Chek1* knockin mouse mice do not have a problem with erythropoiesis.**
**(A, B, C, D)** 21 wild-type (wt) and 15 heterozygous *Chek1* knockin (KD/wt) 45–60-wk-old mice were analyzed for hemoglobin content (A), red blood cell count (B), platelet count, and spleen weight (D). **(E)** Six wt and three KD/wt mice were analyzed for different stages (I–IV) of erythroid development using Ter119/CD71 staining and flow cytometry. Representative flow cytometry plots are seen in Fig S2.

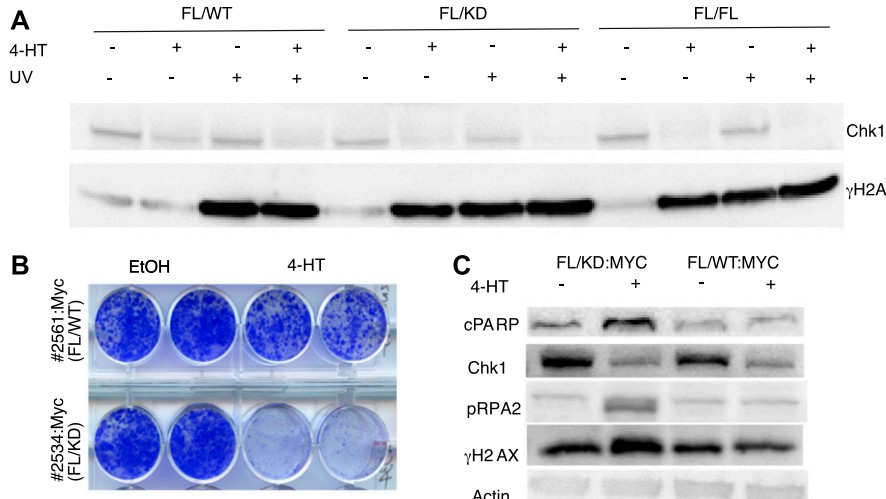

**Figure 4. Chk1 kinase activity is essential to suppress DNA damage signaling activation and for fibroblast viability.**
**(A)** Mouse fibroblasts from CreER mice carrying one floxed Chek1 allele and a wild-type (FL/WT), kinase-dead (FL/KD), or another floxed allele (FL/FL) were treated with vehicle or 4-hydroxytamoxifen (4HT) for 4 d and/or with UVB for 30 s (~700 J/m$^2$) and harvested 2 h after radiation or mock. Immunoblotting was performed for Chk1 and γH2Ax. **(B)** Mouse fibroblast expressing CreER, and a floxed and a kinase-dead or wild-type Chek1 allele were transduced with a Myc-expressing retrovirus. Cells were treated with vehicle or tamoxifen and long-term survival was determined by staining cell monolayers. **(A, C)** Cells manipulated the same as in (A) were analyzed by immunoblotting for indicated markers of apoptosis (PARP), replication stress (threonine 21 phosphorylated-RPA2), and DNA damage (γH2Ax).

replication protein A (p-RPA), induction of DNA damage (γH2AX), and cleavage of PARP, a marker of apoptosis (Fig 4C).

### The Chk1 kinase activity is essential for Myc-induced lymphoma survival

We and others have previously shown that Chk1 inhibitors are very efficacious in killing Myc-induced lymphoma cells. To assess if kinase inhibitor effects can be modeled genetically, we interbred the *CreER;Chek1*$^{Flox/D130A}$ or *CreER;Chek1*$^{Flox/wt}$ with the *λ-Myc* mouse, a transgenic mouse model where c-Myc is under the control of the immunoglobulin *λ* light chain promoter (28). *λ-Myc* mice develop B-cell lymphoma by 100 d of age. To accelerate the process of lymphomagenesis, we interbred to *Cdkn2a*$^{-/-}$ mice. Heterozygous *λ-Myc; Cdkn2a*$^{+/-}$ mice develop tumors at 30–50 d of age and the tumors lose the remaining wild-type *Cdkn2a* allele (29). One kinase-dead allele did not delay this acceleration (Fig 5A). When tumors developed in the offspring, we harvested lymphoma from *λ-Myc; CreER;Chek1*$^{Flox/D130A}$;*Cdkn2a*$^{+/-}$, *λ-Myc;CreER;Chek1*$^{Flox/Flox}$;*Cdkn2a*$^{+/-}$ or *λ-Myc;CreER;Chek1*$^{Flox/wt}$;*Cdkn2a*$^{+/-}$ mice. These tumors could be transplanted into recipient C57BL/6 mice. These mice were monitored for lymphoma manifestation by measuring white blood cells in the blood. When mice had higher than normal blood counts (>12 white cells per nl), each cohort was divided in two and these groups received vehicle or tamoxifen. Mice transplanted with *λ-Myc;CreER; Chek1*$^{Flox/wt}$;*Cdkn2a*$^{+/-}$ lymphoma cells did not respond at all to tamoxifen (Fig 5B), again demonstrating that one allele of *Chek1* is sufficient for lymphoma cells to survive. On the other hand, mice transplanted with lymphoma cells from *λ-Myc;CreER;Chek1*$^{Flox/D130A}$; *Cdkn2a*$^{+/-}$ mice (Fig 5C), and to a lesser extent lymphoma cells from *λ-Myc;CreER;Chek1*$^{Flox/Flox}$;*Cdkn2a*$^{+/-}$ mice (Fig 5D), exhibited a rapid decrease in white blood cell count upon tamoxifen treatment. Despite the acute therapeutic effects in lymphoma by deleting the wild-type alleles of *Chek1*, lymphomas eventually relapsed in the mice. The lymphomas that relapsed did not have any reduced levels of Chk1 protein (Fig S3A). To investigate if this was due to an inefficient Cre-mediated deletion, we genotyped the lymphomas

for the floxed allele and the post-Cre product theoretically generated after Cre-mediated deletion (Fig S3B and C). This analysis demonstrated that the relapsing FL/KD tumors had not undergone Cre deletion of the FL allele, indicating that the relapsing lymphomas were escapees of tamoxifen treatment. This strengthened the view that a kinase-dead allele cannot alone support viability of Myc-induced lymphoma. On the other hand, the FL/wt tumor did show evidence of Cre-mediated deletion, as evidenced by a disappearance of the conditional allele and an emergence of a post-Cre PCR product. These data support the notion that one allele of *Chek1* is sufficient for viability, as long as it is a functional allele. The presence of a post-Cre allele in FL/FL lymphoma suggested that the relapsing lymphoma had only deleted one allele of the conditional alleles and hence remained viable.

### Antitumoral immunity mediated by T cells is not impaired by Chk1 inhibition

The fact that Myc-induced B-lymphoma cells are sensitive to Chk1 inhibition is promising for lymphoma treatment, but recent data suggest that the sensitivity reflects a B-cell lineage sensitivity (30). Important though, other lymphocytes, such as T cells, are essential for antitumor immunity of solid tumors. Therefore, we investigated the impact of Chk1 inhibition on T-cell activity. We investigated the antitumoral function of human T cells during Chk1 inhibition. We cultured melanoma cells labeled with luciferase in the absence or presence of melanoma tumor-infiltrating cells (TILs) and increasing concentrations of the Chk1 inhibitors CCT245737, AZD7762, and GDC-0575. AZD7762 blocked the ability of the TILs to kill the melanoma cells, whereas CCT245737 and GDC-0575 did not inhibit but enhanced the killing of melanoma cells by TILs (Fig 6A). We previously showed that melanoma cells are sensitive to HER2 CAR-T cells (31). When we treated melanoma cells with HER2 CAR-T cells, AZD7762 inhibited the CAR-T cells from killing melanoma cells, whereas CCT245737 and GDC-0575 enhanced the CAR-T–mediated killing of melanoma cells (Fig 6B). The inability of T cells to kill melanoma cells in the presence of AZD7762 correlated with an impaired ability

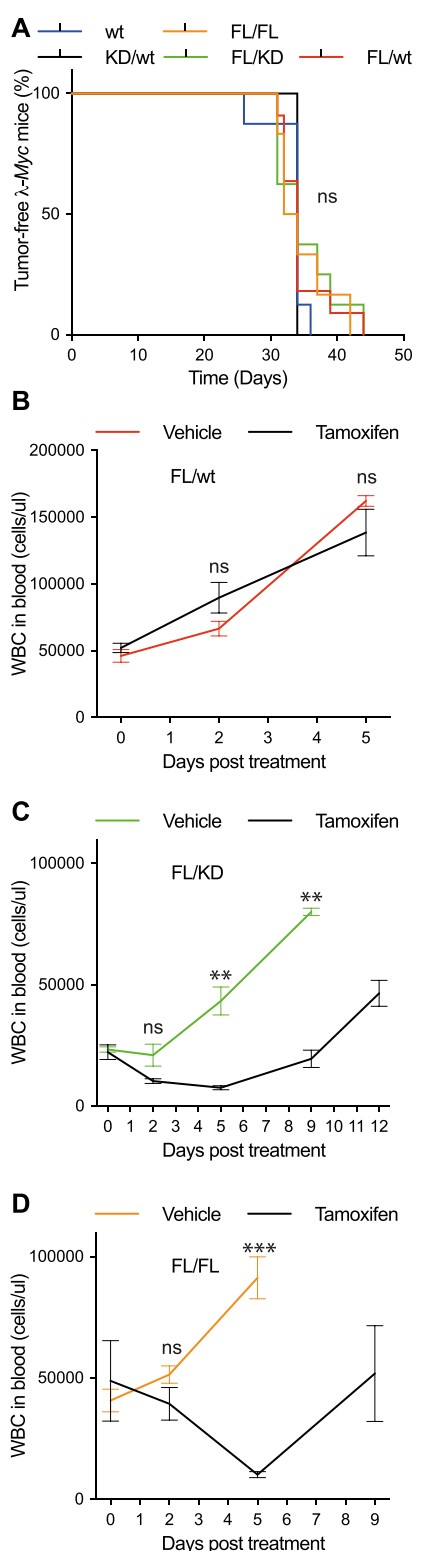

**Figure 5. Expression of only a kinase-dead allele of *Chek1* suppresses lymphoma growth.**
**(A)** Kaplan–Meier survival analysis of λ-*Myc*; *Cdkn2a*$^{+/-}$ mice carrying the indicated *Chek1* genotypes. **(A, B, C, D)** Lymphomas from mice generated from mice in (A) were transplanted into eight C57BL/6 mice per indicated genotype. When white blood counts reached between 20 and 50 × 10³ cells/μl, four mice received vehicle or tamoxifen by i.p. injection. White blood count was

to degranulate (Fig 6C). On the other hand, neither CCT245737, GDC-0575, nor the ATR inhibitor AZ20 negatively affected degranulation. We therefore hypothesized that inhibition of both Chk1 and Chk2 by AZD7762 is less optimal for T-cell combination therapies than inhibiting the ATR/Chk1 axis.

GDC-0575 is an orally available Chk1 inhibitor under clinical development. To test if the inhibitor is compatible with immunotherapy, we used a recently developed model, the PDXv2 model (32), where human melanoma is grown in an immunocompromised mouse transgenic for human IL-2 (hIL2-NOG mice). Infusion of autologous T cells into this mouse model can be used to assess the quality of the patient's TILs, and responses in these mice mimic the responses to adoptive T-cell transfer in the same patients from which the PDXv2 model is developed. We transplanted the same luciferase-labeled melanoma cells used in experiments depicted in Fig 6A–C into hIL2-NOG mice. When tumor growth was confirmed by bioluminescence imaging, we transplanted the tumor-bearing mice with vehicle or TILs. The mice were thereafter treated with vehicle or GDC-0575. As seen in Fig 6D, all mice receiving TILs had a regression of the tumor, irrespective of GDC-0575 co-treatment, whereas GDC-0575 monotherapy was ineffective. We therefore conclude that Chk1 inhibitors could be used in cancer treatment by not negatively impacting tumor immunity.

## Discussion

Here, we present the phenotypes of a mouse where one of the alleles have been mutated to express a variant of Chk1 that has been shown to be kinase-inactive (26). We developed this model to complement our studies of Chk1 kinase inhibition using drugs. The main novel conclusions and implications are as follows: First, heterozygosity of the D130A allele is compatible with viability, tumorigenesis, and even erythropoiesis. This suggests that a kinase-dead form of Chk1 is a nonfunctional protein which does not endow the protein with additional problems. This is in contrast to the kinase-dead forms of both ATM and ATR, where the kinase-dead–expressing mouse models behave differently than the knockout and the heterozygous mice, respectively (33, 34). In these models, the dynamics of the kinase-dead proteins is hindered by their inability to phosphorylate other proteins. We find here that a kinase-dead form of Chk1 does not appear to operate as a dominant negative form, in line with previous cell culture experiments (35). We do however find that there is a difference between the kinase-dead allele and a null allele with respect to erythropoiesis in the heterozygous state. We reason that this is either due to that Chk1 has a kinase activity–independent function in reticulocytes, or that the targeting constructs differ significantly. In our mouse model, a single amino acid change has been introduced, and the selection cassette used during ES targeting has been removed. In Boles et al, where it is claimed that Chk1 is "haploinsufficient" for erythropoiesis, the original *Chek1* knockout mouse was used (27). In this mouse, exons 2–5 of *Chek1* is replaced by a neomycin cassette (neo),

followed until relapse or when palpable tumors required euthanasia of the animals.

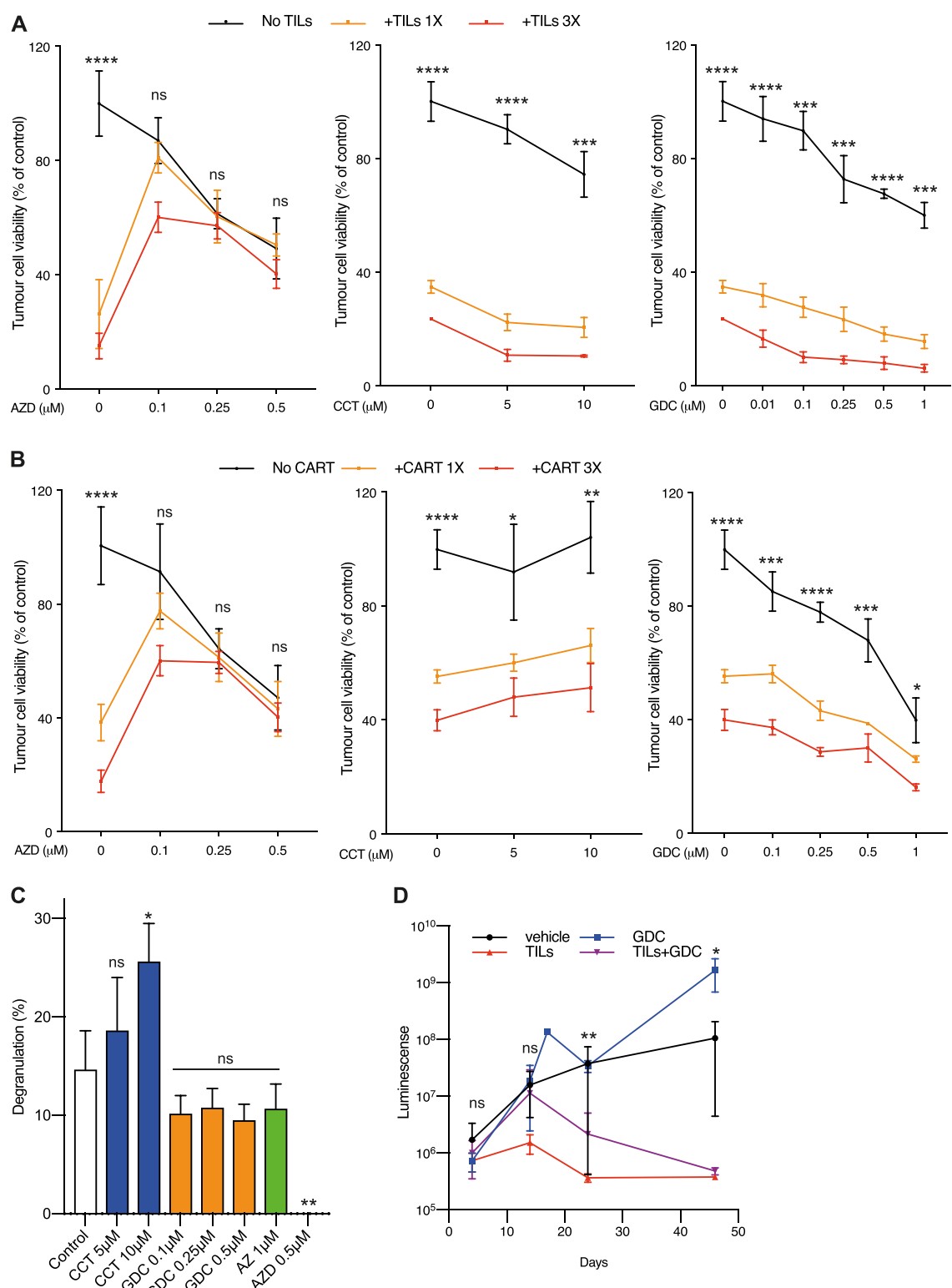

**Figure 6. Chk1 inhibition does not interfere with T-cell–mediated killing of melanoma cells.**
**(A)** 33F8-luciferase expressing melanoma cells were treated with indicated Chk1 inhibitors in the absence or presence of autologous TILs (1:1 or 3:1 TILs vs melanoma cells). Luciferin was added and viability was measured in a 96-well plate luminometer. **(A, B)** The same experimental setup as in (A) but using HER2-directed CAR-T cells instead of TILs. **(C)** 33F8 cells were mixed with autologous TILs in the presence of indicated inhibitors. Surface expression of CD107a as a marker of T-cell degranulation was determined by flow cytometry. **(D)** NOD/SCID/IL2 receptor gamma knockout mice transgenic for human interleukin-2 (hIL2-NOG mice) were transplanted with 33F8-luciferase melanoma cells. When tumors were detectable by IVIS imaging, TILs were injected and mice were dosed in indicated groups of animals (n = 4). Mice were followed with IVIS imaging until tumor size reached the ethics limit (10 mm on the shortest side). Shown are median IVIS signal and standard error of the mean.

which would pose a risk that other genes are affected by the presence of an ectopic promoter driving neomycin. It is of interest to note that the erythropoietin receptor gene *Epor* resides in the vicinity of *Chek1* in the mouse genome. It is thus tempting to speculate that the neo cassette in *Chek1* heterozygous mice could result in a hypomorphic allele of *Epor* resulting in a defect in erythropoiesis in this animal model. Unfortunately, we were unable to confirm this since *Chek1* heterozygous mice was not available at the time when we found that a heterozygous kinase-dead expressing Chk1 mouse did not exhibit any aging-related defects in erythropoiesis. Nevertheless, recently Schuler et al also demonstrated that whereas conditional deletion of both alleles of Chek1 using a Vav-Cre driver impaired hematopoiesis, heterozygous deletion had no impact on hematopoiesis or erythropoiesis (36). These data and ours argue that Chk1 is not haploinsufficient for erythropoiesis.

The second important conclusion we make is that a homozygous *Chek1*^D130A/D130A mouse embryo does not survive after implantation, suggesting that the kinase activity of Chk1 is essential for early embryogenesis. This is in accordance with data using the *Chek1* knockout mouse. We observed that heterozygous *Chek1*^D130A offspring were born in a Mendelian ratio if one parent was heterozygous for the kinase-dead allele, but not if both were. This suggests that heterozygous oocytes or sperms have adverse phenotypes or that there are placental defects, which are only evident when both parents carry a kinase-dead *Chek1* allele. Our acute deletion using tamoxifen of the wild-type allele, while leaving one kinase-dead allele of *Chek1* intact, confirmed that Chk1 regulates its own stability (37) and that some adult cells required Chk1. Indeed, the clinical trials carried out so far have indicated dose-limiting side effects using Chk1 inhibitors. Given that many drugs have side effects, it has been impossible to tie these solely to Chk1 inhibition. Our new animal model offers a unique possibility to study both therapeutic effects and side effects of Chk1 inhibitors.

The third set of conclusions of this study is that we can confirm that Myc overexpressing cells are sensitive to Chk1 inhibition (18). This sensitivity is associated with kinase inhibition and not just a loss of protein. We show that cancer cells are sensitive to Chk1 inhibition but the activity of tumor-killing T cells are not inhibited by Chk1 inhibition. This is an important finding because cancer immunotherapies have become mainstay in many diagnoses, including melanoma, lung cancer, and renal cell carcinoma. Because T cells in our animal model are proliferating because of constitutive IL2 signaling, our data suggest that Chk1 is not as essential for this lineage as for the B-cell lineage. A recent article (30) showed not only that B cells require Chk1 but also that deletion of one allele of *Chek1* delayed lymphomagenesis. We did not observe this dependency which may well represent the differences in B-cell lymphoma model, where we used the *λ-Myc* mouse model and the other researchers used Eμ-*Myc* mice, a model where Myc is driven by an enhancer/promoter element that is activated earlier in the B-cell development than is the λ enhancer/promoter.

Our data showing that anti-tumoral T-cell immunity is not affected by Chk1 inhibitors are in line with recent studies using inhibitors of the upstream kinase ATR, the kinase that activates Chk1. ATR inhibitors synergize with radiotherapy in mouse models by suppressing PD-L1 expression and reducing amounts of T regulatory cells promoting antitumor immunity (2). Trials combining ATR inhibitors and PD-1 inhibitors are on the way (e.g., NCT04095273). Our new data and previous studies showing that ATR and Chk1 inhibitors synergize ((38) and own unpublished data) support that Chk1 inhibitors could be combined with immunotherapy and/or ATR inhibitors in the clinical management of solid tumors such as melanoma.

# Materials and Methods

## Animal ethics

All animal experiments were conducted in accordance with Directive 2010/63/EU and was approved by the regional animal ethics committee of Gothenburg. Approval numbers are #98-2015, #36-2014, and #1183-2018.

## Generation of a mouse expressing a kinase-dead form of Chk1

The *Chek1*^D130A/+ mouse model was made at inGenious Targeting Laboratory. Briefly, the targeting construct to generate the genomic D130A substitution was generated by cloning a 2.45 kb 5′ homology arm and a 5.8 kb 3′ homology arm of *Chek1* from a BAC clone upstream and downstream of a FRT-NEO-FRT cassette, respectively. The 3′ homology arm contained an A→C substitution compared with the wild-type sequence. The targeting construct was electroporated into C57BL/6 ES cells and 300 clones were picked after selection. Five of these were selected after PCR confirmation for verification by sequencing of the mutation, Southern blot, and karyotyping. All passed with the correct mutation, only one integration and ≥89% euploidy. Two of the five clones were used for blastocyst injection into Balb/c blastocysts, called line 1 and line 2. Line 1 generated three high and two medium percentage chimeras. Lines 2 generated two high and one medium percentage chimeras. The chimeric mice were bred to B6 FLP mice, to remove the neo cassette. Offspring were genotyped to confirm loss of neo cassette and a region containing the D130A mutation was amplified by PCR and sequenced. These were sent to the University of Gothenburg and were backcrossed to C57BL/6 once and genotyped to confirm neo deletion, before interbreeding heterozygous mice. Neither heterozygous mice derived blastocyst injection ES cell line 1 nor from line 2 blastocyst injections were able to generate viable homozygous offspring (Fig 2 and data not shown). For the rest of the experiments, we used only mice and cells derived from line 1.

To be able to assess the effects of expression of a kinase-dead form of Chk1, we interbred the *Chek1*^D130A/+ mouse to a conditional floxed *Chek1* mouse (10) and to a transgenic mouse expressing CreER both from The Jackson Laboratories. The latter mouse expresses a Cre recombinase fused to the ligand binding domain of the estrogen receptor. Treatment of mice or cells expressing CreER results in an ability of Cre to translocate to the nucleus and to recombine DNA elements flanked by lox P sites. The compound mice were also bred to a *λ-Myc* transgenic mouse (The European Mouse Mutant Archive repository) and to *Cdkn2a* knockout mouse (Jackson Laboratories).

## Cell culture

MEFs were cultured in DMEM with stable glutamate (Gibco) supplemented with 10% heat-inactivated FBS, gentamycin, sodium pyruvate, and β-mercaptoethanol. TILs (32) and HER2-CAR-T cells (31) were cultured in RPMI with stable glutamate, supplemented with 10% human serum, gentamycin, and 6,000 units of human IL-2 (Pepro Tech). 33F8-Luc cells (32) were cultured in RPMI supplemented with 10% heat-inactivated FBS and gentamycin. All cells were tested for mycoplasma by PCR and were cultured at 37°C and 5% $CO_2$.

## Inhibitors

ATR inhibitor AZ20 was purchased from MedChem express, whereas Chk1 inhibitors CCT245737, AZD7762, and GDC-0575 were purchased from Selleckchem. Inhibitors were dissolved in DMSO at 10 mM and stored at −20°C or −80°C.

## Cell viability

For cell viability measurements, cells were cultured in 96-well plates and metabolic activity was measured using Cell-Titer-Glo assay or CellTiter-Fluor Cell Viability Assay (Promega), as per the manufacturer's protocol, in GloMax plate reader. Killing assay was performed in 96-well plates. For killing assay, 33F8 cells expressing Luciferase were seeded in 96-well plates, and 33F8 TILs were added to them at varying ratios, in the presence or absence of various Chk1 inhibitors. Luciferase assay was carried out 24 and 48 h post-treatment to assess viability of the tumor cells.

## Immunoblotting

Cell pellets were lysed in Arf lysis buffer as described earlier (39). In all, 50 μg of proteins were resolved on 4–20% Bio-Rad Mini-PROTEAN TGX gels and then transferred onto nitrocellulose membrane, following which, the membrane was blocked in 5% BSA or milk. Blocked membranes were then blotted with antibodies directed against the following proteins: p-Chk1, cleaved PARP (Cell Signaling Technology), Chk1 (Santa Cruz Biotechnology), Actin (Sigma-Aldrich), or phosphorylated–histone–H2AX (S139, γH2AX; Merck Millipore).

## Flow cytometry analysis

For cell cycle analysis, around one million cells per ml were lysed in Vindelo's solution (20 mmol/l Tris, 100 mmol/l NaCl, 1 μg/ml 7-AAD, 20 μg/ml RNase, and 0.1% NP40) for 30 min at 37 degrees. Stained nuclei were analyzed using a BD Accuri C6 flow cytometer. Apoptosis was measured on a logarithmic scale on FL3 channel, whereas cell cycle of viable cells was analyzed on linear scale.

For immunophenotyping of erythrocyte lineage development, bone marrows from wild-type or $Chek1^{D130A/wt}$ mice (n = 6 or 3) were incubated with PE-conjugated CD71 and APC-conjugated Ter119 antibodies for 15 min in the refrigerator. Cells were washed and analyzed in a BD Accuri C6 flow cytometer.

# Supplementary Information

# Acknowledgements

We thank Sofia Stenqvist and Mona Svedman for animal care, the personnel at Experimental Biomedicine for animal husbandry, and Carina Karlsson and Samuel Alsén for technical assistance. Funding: This study was made possible by generous research support from Knut and Alice Wallenberg Foundation, Familjen Erling Persson Foundation, Cancerfonden, Vetenskapsrådet, The Sjöberg Foundation, Lion's Cancer Foundation Western Sweden, Sahlgrenska Hospital (ALF grant), and BioCARE governmental strategic funds for cancer research to University of Gothenburg.

## Author Contributions

SV Muralidharan: data curation, formal analysis, and writing—review and editing.
LM Nilsson: data curation, formal analysis, and writing—review and editing.
MF Lindberg: data curation, formal analysis, and writing—review and editing.
JA Nilsson: conceptualization, formal analysis, supervision, funding acquisition, visualization, project administration, and writing—original draft, review, and editing.

## Conflict of Interest Statement

The authors declare that they have no conflict of interest.

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
