## [Reviewer comments · Life Science Alliance]

Life Science Alliance

A novel mouse models shows that the Chk1 kinase activity is essential for embryos and cancer cells

Somsundar Muralidharan, Lisa Nilsson, Mattias Lindberg, and Jonas Nilsson

DOI: <https://doi.org/10.26508/lsa.202000671>

Corresponding author(s): Jonas Nilsson, University of Gothenburg

Review Timeline:	Submission Date:	2020-02-07
	Editorial Decision:	2020-03-05
	Revision Received:	2020-06-04
	Editorial Decision:	2020-06-05
	Revision Received:	2020-06-12
	Accepted:	2020-06-12

Transaction Report:

March 5, 2020

Re: Life Science Alliance manuscript #LSA-2020-00671-T

Jonas Andrej Nilsson
University of Gothenburg
Sahlgrenska Cancer Center, Department of Surgery, Institute of Clinical Sciences, University of Gothenburg
Sahlgrenska Cancer Center
Medicinaregatan 1G, plan6
Gothenburg 40530
Sweden

Dear Dr. Nilsson,

Thank you for submitting your manuscript entitled "The kinase activity of Chk1 is essential for mouse embryos and cancer cells" to Life Science Alliance. The manuscript was assessed by expert reviewers, whose comments are appended to this letter.

As you will see, the reviewers think that your findings will be of significant value for the field. The reviewers provide constructive input on how to further strengthen your manuscript, and we would thus like to invite you to submit a revised version of your manuscript to us, addressing the comments made by the reviewers. This seems rather straightforward, but please do get in touch in case you would like to discuss individual revision points further.

Thank you for this interesting contribution to Life Science Alliance. We are looking forward to receiving your revised manuscript.

Sincerely,

B. MANUSCRIPT ORGANIZATION AND FORMATTING:

Reviewer #1 (Comments to the Authors (Required)):

This is a solid study that describes the generation and characterization of a Chk1 kinase-dead mouse. Heterozygous mice are viable and show no signs of anemia. Chk1 kinase activity is essential for mouse and myc-induced lymphoma survival. Finally Chk1 inhibition does not interfere

with T-cell mediated killing of melanoma cells. This latter piece of the manuscript is limited as the authors do not address whether the T cells are dividing in the presence of the Chk1 inhibitor. Nevertheless the study is strong and has significant clinical relevance.

The third line of the abstract makes no sense and there are many places where the text could be improved to clarify what the authors mean.

Reviewer #2 (Comments to the Authors (Required)):

In this manuscript from Nilsson laboratory, authors generate a novel mouse model harboring a kinase-dead Chk1 allele. They show that while live-born heterozygous Chk1-kd mice appear normal, a homozygous Chk1-kd/kd mice died shortly after E3.5, suggesting that kinase activity of Chk1 is essential for embryogenesis and viability. Interestingly, kinase-dead Chk1 is able to suppress cancer growth, in a Myc lymphoma mouse model. But the erythropoietin development and the anti-tumoral immunity mediated by T cells are not impaired by the Chk1 kinase inhibition. Some concerns need to be addressed before considering the paper for publication.

- 1) It would be helpful to verify the level of CHK1-kd protein and the DNA damage induced CHK1 activity by Chk1 D130A (het or homo if available). It could be done by measuring cdc25 levels in Fig 4c cells.
- 2) This might also explain why the Chk1+/kd were under-represented in the pups. Is there a statistically significant loss of +/kd? A Fisher's exact test would be helpful.
- 3) Although it is a negative phenotype, it would be helpful to include a histology of bone marrow or spleen or a flow cytometry analysis of red blood cell development (e.g., CD71-Ter11) in figure 3.
- 4) In figure 4c, could the authors clarify which residue of RPA was phosphorylated in western blot? Since RPA can be phosphorylated on several residues (T21, S4/S8, S33).

Minor comments.

- 1) Interestingly, Chk1-kd expression suppresses lymphoma growth in a lambda-MYC mouse, a model for Burkitt lymphoma. Authors say that lymphoma eventually relapse in the mice analyzed without providing any further details. Did they analyze by flow cytometry or histology the tumors developed? Do they resemble the tumors in the lambda-myc mice with WT CHK1? Besides WBC count, a more detailed analysis of the tumors in the different genotypes would be helpful, if possible.
- 2) Figures 5 and 6 miss that statistical significance statement and p values.
- 3) Supplementary Figure 1 title should be "...does NOT impact the growth of mouse..."
- 4) The manuscript could be benefited from carefully proof-reading and revision on grammar, sentences structure and composition.

Reviewer #3 (Comments to the Authors (Required)):

This is a valuable contribution dissecting the role of the kinase activity of CHK1 from potential scaffolding functions of the protein. In essence kinase dead mutant versions of CHK1 recapitulate the phenotype of CHK1 deletion during embryogenesis. Moreover, the study shows that kinase activity is critical for tumor cell survival in a MYC driven lymphoma model. Additionally, the authors provide evidence that a certain window of opportunity exists for the use of CHK1i also when combined with immunotherapy. I think it is critical to share all the information with the community. What I do miss though is a more critical discussion, related to the effects caused by the expression

of one mutant allele, which is expected to mimic CHK1 haploinsufficiency. In ref 30, the authors actually also report that loss of one allele delays MYC driven lymphomagenesis. Differences between the model systems used should be discussed.

Similarly, the discussion about the effects of haploinsufficiency on erythropoiesis could be discussed in light of the findings published by Schuler et al in EMBOR 2019, using Vav-CRE, where heterozygous mice also show no impairment of hematopoiesis, including erythrocyte development. Finally, the fact that kinase inhibitors lead to protein degradation due to impaired autophosphorylation on Ser 296, triggering ubiquitination of CHK1, as recently published in JCB, should also be discussed.

Reviewer #1 (Comments to the Authors (Required)):

This is a solid study that describes the generation and characterization of a Chk1 kinase-dead mouse. Heterozygous mice are viable and show no signs of anemia. Chk1 kinase activity is essential for mouse and myc-induced lymphoma survival. Finally Chk1 inhibition does not interfere with T-cell mediated killing of melanoma cells. This latter piece of the manuscript is limited as the authors do not address whether the T cells are dividing in the presence of the Chk1 inhibitor. Nevertheless the study is strong and has significant clinical relevance.

We thank the reviewer for the kind words.

The third line of the abstract makes no sense and there are many places where the text could be improved to clarify what the authors mean.

We have altered the third sentence for clarity and gone through the text for improvement.

Reviewer #2 (Comments to the Authors (Required)):

In this manuscript from Nilsson laboratory, authors generate a novel mouse model harboring a kinase-dead Chk1 allele. They show that while live-born heterozygous Chk1-kd mice appear normal, a homozygous Chk1-kd/kd mice died shortly after E3.5, suggesting that kinase activity of Chk1 is essential for embryogenesis and viability. Interestingly, kinase-dead Chk1 is able to suppress cancer growth, in a Myc lymphoma mouse model. But the erythropoietin development and the anti-tumoral immunity mediated by T cells are not impaired by the Chk1 kinase inhibition. Some concerns need to be addressed before considering the paper for publication.

1) It would be helpful to verify the level of CHK1-kd protein and the DNA damage induced CHK1 activity by Chk1 D130A (het or homo if available). It could be done by measuring cdc25 levels in Fig 4c cells.

We have attempted to do Western blot of Cdc25c but did not manage to detect the protein. However, we have evidence that the Chk1 D130A allele causes DNA damage in the absence of a wildtype allele as evidenced by γ H2ax immunoblots (Figure 4A and 4C). However, KD/wt cells do not induce DNA damage, arguing against a dominant negative action of the kinase-dead allele, as been demonstrated in cell line experiments (Gatei et al., 2003 JBC).

We tried to quantify the levels of the D130A Chk1 form but unfortunately the peptides containing the D130 or A130 residues were not detectable in our proteomics experiment. Nevertheless, Figure 4A and 4C demonstrates that when the wt allele is removed genetically, the protein levels of Chk1 is decreased, confirming that Chk1 regulates its own stability (Michelena et al., JCB 2019). We feel that we have provided the evidence needed to demonstrate that the D130A is inactive in its function, which is expected based on the original paper describing this mutation (Sanchez et al., 1997 Science).

We thank the reviewer for allowing us to discuss these issues and we have added the references above in the discussion of manuscript to clarify our conclusions.

2) This might also explain why the Chk1+/kd were under-represented in the pups. Is there a statistic significant loss of +/kd? A fisher's exact test would be helpful.

As mentioned above, there is no evidence of any dominant-negative effects of the D130A allele. When heterozygous KD mice are interbred with wt C57BL/6 mice, a Mendelian distribution. We have added in this data as a new figure 2e. We cannot explain why heterozygous pups are lost in the het x het breeding but when doing a Fisher test it was borderline significant ($p=0.06$). We therefore added thirty offspring from additional het x het breeding of line 2 (see Methods) and again the trend was the same. Combining all data (updated in the Figure 2d) resulted in a significant loss of heterozygous offspring. We added in the data from the Fisher's test and thank the reviewer for this suggestion of an analysis.

3) Although it is a negative phenotype, it would be helpful to include a histology of bone marrow or spleen or a flow cytometry analyses of red blood cell development (e.g., CD71-Ter11) in figure 3.

The data in Figure 3 was indeed based on flow cytometry. We have added representatives plots in a new Suppl figure 2. We have also added in a description of the methodology which was unfortunately missed in the Methods section.

4) In figure 4c, could the authors clarify which residue of RPA was phosphorylated in western blot? Since RPA can be phosphorylated on several residues (T21, S4/S8, S33).

We have added in the information.

Minor comments.

1) Interestingly, Chk1-kd expression suppresses lymphoma growth in a lambda-MYC mouse, a model for Burkitt lymphoma. Authors say that lymphoma eventually relapse in the mice analyzed without providing any further details. Did they analyze by flow cytometry or histology the tumors developed? Do they resemble the tumors in the lambda-myc mice with WT CHK1? Besides WBC count, a more detailed analyses of the tumors in the different genotypes would be helpful, if possible.

We have now analyzed tumors representative of the three λ -MYC; *Chk1* genotypes (FL/wt, FL/KD and FL/FL) that eventually reappeared in the mice after treatment with tamoxifen. Our analyses show that the lymphomas that relapsed did not have any reduced levels of Chk1 protein (new supplementary figure S3A). To investigate if this was due to an inefficient Cre-mediated deletion we genotyped the lymphomas for the floxed allele and the post-Cre product theoretically generated after Cre-mediated deletion (new supplementary figure S3B-C). This analysis demonstrates that the relapsing FL/KD tumors have not undergone Cre deletion of the FL allele indicating that the relapsing lymphoma is an escapee of tamoxifen treatment. This strengthen the view that a kinase dead allele cannot alone support viability of Myc-induced lymphoma. On the other hand, the FL/wt tumors do show evidence of Cre-mediated deletion, supporting the notion that one allele of *Chk1* is sufficient for viability, as long as it is a functional allele. The presence of a post-cre allele in FL/FL lymphoma suggest that the lymphoma arising only deleted one allele of the floxed allele and hence remained viable.

We thank the reviewer for this comment as it has resulted in an interesting analysis that strengthen the view that Chk1 is essential for Myc-induced lymphoma. We have added in a section in the results.

2) Figures 5 and 6 misses that statistical significance statement and p values.

We have added in statistical statements and p-values.

3) Supplementary Figure 1 title should be "...does NOT impact the growth of mouse..."

We thank the reviewer for spotting this mistake. We have now changed it.

4) The manuscript could be benefited from carefully proof-reading and revision on grammar, sentences structure and composition.

We have read through the manuscript an altered a few sentences for clarity.

Reviewer #3 (Comments to the Authors (Required)):

This is a valuable contribution dissecting the role of the kinase activity of CHK1 from potential scaffolding functions of the protein. In essence kinase dead mutant versions of CHK1 recapitulate the phenotype of CHK1 deletion during embryogenesis. Moreover, the study shows that kinase activity is critical for tumor cell survival in a MYC driven lymphoma model. Additionally, the authors provide evidence that a certain window of opportunity exists for the use of CHK1i also when combined with immunotherapy. I think it is critical to share all the information with the community.

What I do miss though is a more critical discussion, related to the effects caused by the expression of one mutant allele, which is expected to mimic CHK1 haploinsufficiency. In ref 30, the authors actually also report that loss of one allele delays MYC driven lymphomagenesis. Differences between the model systems used should be discussed.

We have discussed the difference of models used in our study and reference 30 in a new section of the discussion. We thank the reviewer for this point.

Similarly, the discussion about the effects of haploinsufficiency on erythropoiesis could be discussed in light of the findings published by Schuler et al in EMBOR 2019, using Vav-CRE, where heterozygous mice also show no impairment of hematopoiesis, including erythrocyte development.

We have added in this very important paper in the discussion. It supports our conclusions so we are grateful that the reviewer mentioned it.

Finally, the fact that kinase inhibitors lead to protein degradation due to impaired autophosphorylation on Ser 296, triggering ubiquitination of CHK1, as recently published in JCB, should also be discussed.

We have added in this very important paper in the discussion. It supports our conclusions so we are grateful that the reviewer mentioned it.

June 5, 2020

RE: Life Science Alliance Manuscript #LSA-2020-00671-TR

Prof. Jonas Andrej Nilsson
University of Gothenburg
Sahlgrenska Cancer Center, Department of Surgery, Institute of Clinical Sciences, University of Gothenburg
Sahlgrenska Cancer Center
Medicinaregatan 1G, plan6
Gothenburg 40530
Sweden

Dear Dr. Nilsson,

Thank you for submitting your revised manuscript entitled "A novel mouse models shows that the Chk1 kinase activity is essential for embryos and cancer cells". We would be happy to publish your paper in Life Science Alliance pending final revisions necessary to meet our formatting guidelines.

- please have corresponding authors add their ORCID ID - you should have received instructions on how to do so
- please add all required information in our system - the category and article type are missing
- please add callouts in your manuscript text for Figure 2E and Figure S2
- please list 10 authors et al in your reference list
- During a standard image analysis, we note that the resolution of the image for the upper blot in figure 1A is poor resolution compared to the other panels in the figure. In line with journal policies, we are requesting the unmodified source data for all the blots in Figure 1

A. FINAL FILES:

- An editable version of the final text (.DOC or .DOCX) is needed for copyediting (no PDFs).

- High-resolution figure, supplementary figure and video files uploaded as individual files: See our detailed guidelines for preparing your production-ready images, <http://www.life-science->

alliance.org/authors

B. MANUSCRIPT ORGANIZATION AND FORMATTING:

Sincerely,

Reilly Lorenz
Editorial Office Life Science Alliance
Meyerhofstr. 1
69117 Heidelberg, Germany
t +49 6221 8891 414
e contact@life-science-alliance.org
www.life-science-alliance.org

Re: Life Science Alliance manuscript #LSA-2020-00671-T

Thank you for accepting our manuscript. We got a few editorial questions – here's how we respond to them:

-please have corresponding authors add their ORCID ID - you should have received instructions on how to do so

Those that have an ORCID ID have added it.

-please add all required information in our system - the category and article type are missing

I added this.

-please add callouts in your manuscript text for Figure 2E and Figure S2

I have added this, thanks for spotting it!

-please list 10 authors et al in your reference list

I changed this.

-During a standard image analysis, we note that the resolution of the image for the upper blot in figure 1A is poor resolution compared to the other panels in the figure. In line with journal policies, we are requesting the unmodified source data for all the blots in Figure 1

I have exchanged Figure 1A Chk1 blot for the same blot in higher resolution. I also add a zip-file with all uncropped images, directly from the LAS machine.

I hope all is order now for sending the manuscript to the printer.

Best regards and Stay safe!
Jonas Nilsson

June 12, 2020

RE: Life Science Alliance Manuscript #LSA-2020-00671-TRR

Prof. Jonas Andrej Nilsson
University of Gothenburg
Sahlgrenska Cancer Center, Department of Surgery, Institute of Clinical Sciences, University of Gothenburg
Sahlgrenska Cancer Center
Medicinaregatan 1G, plan6
Gothenburg 40530
Sweden

Dear Dr. Nilsson,

Thank you for submitting your Research Article entitled "A novel mouse models shows that the Chk1 kinase activity is essential for embryos and cancer cells". It is a pleasure to let you know that your manuscript is now accepted for publication in Life Science Alliance. Congratulations on this interesting work.

DISTRIBUTION OF MATERIALS:

Again, congratulations on a very nice paper. I hope you found the review process to be constructive

and are pleased with how the manuscript was handled editorially. We look forward to future exciting submissions from your lab.

Sincerely,

Reilly Lorenz
Editorial Office Life Science Alliance
Meyerhofstr. 1
69117 Heidelberg, Germany
t +49 6221 8891 414
e contact@life-science-alliance.org
www.life-science-alliance.org